# Seeded LoRA: Collaborative Fine-Tuning Through Seed Initialization of Adapters

## Abstract

Parameter-Efficient Fine-Tuning (PEFT) methods facilitate the cost-effective adaptation of pretrained language models to specific tasks and domains. These methods have enabled the open-source community to develop thousands of specialized models tailored to various domains and tasks. Collaborative Fine-Tuning (CoFT) is the paradigm that seeks to merge these specialized models into a single model – often a routed Mixture-of-Expert (MoE) model – to achieve better generalization across domains and tasks. However, current CoFT models require a post-merge fine-tuning stage to successfully combine existing models, making CoFT approaches inaccessible to users who lack fine-tuning expertise. In this work, we introduce Seeded LoRA, a novel CoFT approach that does not require post-merge fine-tuning thus enabling plug-and-play PEFT adapter merging. Seeded LoRA significantly outperforms LoRA and MoE LoRA (MoLoRA) approaches, improving by an average of 7 percentage points across a battery of 16 zero-shot tasks and we find that the main benefit from Seeded LoRA comes from mitigating task interference during finetuning. Seeded LoRA works by initializing a model before fine-tuning using a generic seed expert low-rank adapter which was finetuned on a small random subset of the finetuning data such that subsequent fine-tuning runs are initialized in the same optimization subspace. This process enables the integration of any combination of independently fine-tuned models through simple averaging of expert adapter outputs. We show that averaging, or routing with assigning equal probability weights to each expert, is equivalent to grouped convolution, explaining its effectiveness. Additionally, we study subtle routing failures in post-merge fine-tuning and highlight that Seeded LoRA can alleviate most routing failures, making it a suitable base method for future routed CoFT approaches.

## 1 Introduction

Fine-tuning pretrained Large Language Models (LLMs) to follow the instructions of a user(Wei et al., 2022) – also known as post-training – is a key step in developing interactive chatbots. Parameter-Efficient Fine-Tuning (PEFT)(Hu et al., 2021; Liu et al., 2022; Li & Liang, 2021; Zadouri et al., 2023) methods like Low-Rank Adaptation (LoRA)(Hu et al., 2021) have enabled the creation of numerous domain-specific models(Wolf et al., 2019; Mangrulkar et al., 2022). However, adding a capability to augment an existing model, for example, adding code generation to a model trained on mostly English data, traditionally requires re-training with new data mixes, which incurs high computational costs and requires domain-specific expertise for dataset generation and fine-tuning.

Collaborative fine-tuning (CoFT) aims to extend the capabilities of an existing model by *merging* it with other existing fine-tuned models, thus reusing the expertise and computational resources that went into creating already existing models. However, current CoFT strategies often necessitate post-merge fine-tuning to enable successful use of existing PEFT models (Muqeeth et al., 2024; Dou et al., 2024; Zadouri et al., 2023).

**In this paper,** we introduce Seeded LoRA, a CoFT approach that does not require post-merge fine-tuning. Seeded LoRA works by initializing a model before fine-tuning by using a generic seed expert low-rank adapter (LoRA), which was on a random subset of the finetuning data such that subsequent fine-tuning runs are initialized in the same optimization subspace. While this work

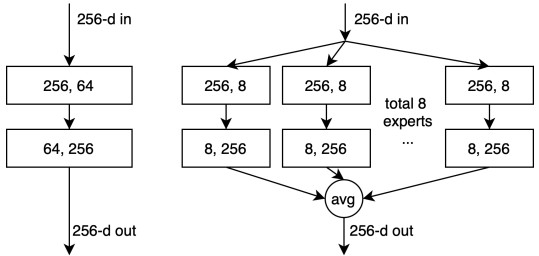

Figure 1: **Left:** LoRA (Hu et al., 2021) adapter with a rank of 64. **Right:** Seeded LoRA with 8 experts, each one with a rank of 8. Both adapters have the same parameter count.

applies this approach for the first time to LoRA modules, it has been shown that in other settings this procedure leads to linear mode connectivity (Frankle et al., 2020), such that existing models can be merged by simple averaging (Li et al., 2022; Wortsman et al., 2022; Ilharco et al., 2022). Our results for LoRA are consistent with these finding as we show we do not require complicated post-merge fine-tuning of a router, but instead are able to add capabilities to a model by simply adding diverse specialized LoRA modules (coding, math, etc.) and averaging their hidden states.

Our findings demonstrate that Seeded LoRA provides state-of-the-art CoFT performance, on par with more complex routed models while being a much simpler approach that does not require post-merge fine-tuning. We show that the main benefit from Seeded LoRA comes from mitigating interference between finetuning tasks(Luo et al., 2023).

While it may first appear limiting that the seed adapter has to be finetuned on about 10% of data, it has been shown that initialization into the same optimization subspace is more important than the exact data used(Gururangan et al., 2023; Li et al., 2022; Frankle et al., 2019). As such, the Seeded LoRA approach remains flexible with respect to the data used to create the seed expert and the data used during subsequent finetuning of experts from the seed expert.

We evaluated Seeded LoRA on 16 different zero-shot tasks and find that compared to LoRA and Mixture of LoRA adapters (MoLoRA) baselines improves the average accuracy from 44.1% (LoRA) and 45.6% (MoLoRA) to 52.5% – a very significant increase in overall performance. We show that this difference in performance stems mostly from arithmetic tasks that have significant task interference – it is usually difficult to do well on these tasks when mixed with other post-training datasets.

Furthermore, we show in our analysis that most routing techniques for CoFT have subtle failure modes that lead to poor performance. Seeded LoRA initialization can be used to overcome these failure modes.

In summary, seeded LoRA facilitates the straightforward combination of arbitrary LoRA models to extend LLM capabilities without additional post-merge fine-tuning, thus accelerating the progress within the open-source community.

## 2 BACKGROUND

**Low-Rank Adaptation (LoRA)** freezes the pretrained parameters of a model and adds only a small set of trainable parameters called low-rank adapters (Hu et al., 2021). This decreases the memory required for fine-tuning by a factor of roughly 6x through reduced memory requirements for gradients and optimizers states. Given a pretrained weight matrix $\mathbf{W}_0 \in \mathbb{R}^{h \times o}$ and intermediate token activation $\mathbf{x} \in \mathbb{R}^h$, LoRA adds a low-rank projection to the outputs of the layer as follows:

$$\mathbf{y} = \mathbf{x}\mathbf{W}_0 + \mathbf{x}\mathbf{A}\mathbf{B} \tag{1}$$

where $A \in \mathbb{R}^{h \times r}$, $B \in \mathbb{R}^{r \times o}$ and $r$ is the rank. Only the weights $\mathbf{A}$, and $\mathbf{B}$ are updated during fine-tuning.

**Collaborative Fine-Tuning (CoFT)** Traditional fine-tuning of a LLM often results in a model with static capabilities, where introducing new functionalities, such as mathematical problem-solving, might erase previously learned skills due to catastrophic forgetting (Goodfellow et al., 2013). Typically, enhancing a model's capabilities involves retraining it from scratch with a comprehensive dataset encompassing both old and new domains – a process that is not only computationally intensive but also demands access to all previous data and domain-specific fine-tuning expertise. Collaborative Fine-Tuning (CoFT) addresses these limitations and extends the capabilities of existing model *without necessitating retraining*. For instance, incorporating math skills into a model could be achieved by *merging* it with another model specifically fine-tuned for mathematics. Among various integration methods, the most common involves deploying a router to manage the interaction between these specialized models (Muqeeth et al., 2024).

**CoFT vs Federated Learning** In federated learning (Lim et al., 2020), individual models are locally trained on edge devices and then the local model parameters are merged on a central server. This allows the training on private data on the edge device while preserving the privacy of edge device users by exchanging parameters but not their user data. While CoFT is similar to federated learning in that parameters are merged into a final model it has some core differences. In CoFT, privacy is not a major consideration while final performance of the merged model is critical; CoFT usually allows only for a single exchange of model parameters and not successive updates; large models are used that cannot be executed on edge devices; large models also introduce new challenges that do not exist at the small scale (Dettmers et al., 2022). CoFT is mainly useful for communities of independent developers that do not have the resources of large organizations and which might act independently with few resources. In summary, while the technical problems are similar, federated learning approaches are usually not suitable CoFT solutions and vice versa due to different emphasis of model scale, single step merging, and the importance of final model performance.

**Mixture-of-Adapters (MoA) methods** such as MoLoRA, SIRA, and LoRAMoE (Zadouri et al., 2023; Zhu et al., 2023; Dou et al., 2024) typically learn a set of experts $E_1, ..., E_n$, where each expert $E_i$ is a LoRA adapter, and a router network $R$ that is parameterized by a dense layer with weights $W_R \in \mathbb{R}^{h \times n}$. The router network takes intermediate token activations $x$ as input and generates the gating scores $s_1, ..., s_n$ for each token that is then used to combine the experts outputs in a weighted sum of all experts (soft mixture) (Masoudnia & Ebrahimpour, 2014) or the experts with the top-k probability (sparse mixture) (Lepikhin et al., 2020; Shazeer et al., 2017):

$$\mathbf{s}_i = R(\mathbf{x})_i = \text{softmax}(\mathbf{W}_R^T \mathbf{x}) \quad \text{(Router)}$$

$$\mathbf{y} = \sum_{i=1}^{n} \mathbf{s}_i \cdot E_i(\mathbf{x}) \quad \text{(MoA Layer)} \tag{2}$$

**Optimization Landscape: Dynamics and Initialization** A key result we build on is that neural networks that are being trained from a random parameter initialization quickly settle into an optimization subspace characterized by the principle eigenvalues of the Hessian (Ghorbani et al., 2019; Gur-Ari et al., 2018; Frankle et al., 2019). Once this subspace is entered the principal optimization directions remain largely fixed for the rest of the training (Frankle et al., 2019; Ghorbani et al., 2019; Gur-Ari et al., 2018) and exhibit linear mode connectivity(Frankle et al., 2020). This means two neural networks with different random initialization may have different optimization subspaces, but if two neural networks are trained from an initialization that has settled into an optimization subspace, the two neural networks will remain in that subspace even if trained on different data (Frankle et al., 2020; Li et al., 2022; Gururangan et al., 2023). If two neural networks exhibit linear mode connectivity, they can be merged by a simple or weighted average (Li et al., 2022; Gururangan et al., 2023; Wortsman et al., 2022; Ilharco et al., 2022). Our main contribution is to exploit this property to enable CoFT that does not require any post-merge fine-tuning.

## 3 SEEDED LORA

Seeded LoRA (Figure 2) builds upon the foundation laid by LoRA (Hu et al., 2021) and introduces key improvements over other mixture of adapters methods. The main innovation is to train a seed

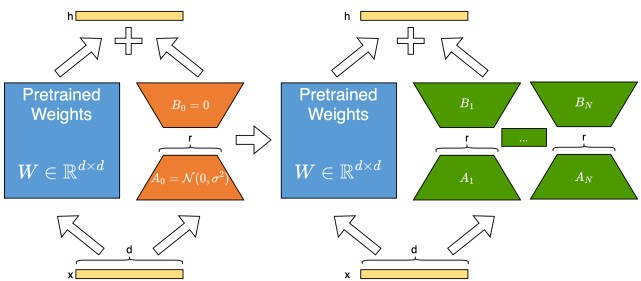

Figure 2: **Left:** Stage I: Seed Adapter training. **Right:** Stage II: Each adapter is initialized from the Seed Adapter as starting point but is trained on different data. As such, each adapter acts as an Expert in this MoA model. Inputs are sent to every expert, and the outputs are averaged and added to the pretrained model output.

expert on a random subset of the data and to use this seed adapter as an initialization in subsequent training of experts. With this approach we can successfully specialize adapters into experts while being able to simply average the outputs of multiple experts for improved performance. This stands in stark contrast to other approaches that use complex dynamic routing mechanisms to combine expert adapters. As we show in Section 6, we find that more complex routing strategies have no advantages compared to simple averaging when we initialize adapters with a seeded expert. Our approach is formulated with the following equations and finetuning process:

$$\mathbf{y} = \mathbf{x}\mathbf{W}_0 + \underbrace{\frac{1}{N}\sum_{i=1}^{N}\mathbf{x}\mathbf{A}_i\mathbf{B}_i}_{\text{Seeded LoRA update}} \tag{3}$$

Here, $\mathbf{W}_0$ represents the base model parameters, $\mathbf{x}$ is the input, and $\mathbf{B}_i$ and $\mathbf{A}_i$ are the $i-th$ LoRA adapter.

**1. *Seed Expert* Training**  A single *seed expert* model is trained on a random subset of the data – we use 10% but as little as 5% can be sufficient (Frankle et al., 2020; Gururangan et al., 2023). This ensures all subsequent experts share a common optimization space that exhibits linear mode connectivity (Frankle et al., 2020). Previous work has shown that the choice of seed data is not as important as having a common optimization space(Gururangan et al., 2023).

**2. *Dataset-Specific Expert* fine-tuning**  Each expert is then fine-tuned independently on a particular domain-specific dataset, such as coding or math datasets. This simulates collaborative fine-tuning (CoFT), that is, independent fine-tuning runs of open-source models from the seed expert.

**3. Merging Experts**  Finally, Seeded LoRA incorporates all fine-tuned adapters through a simple average of the output hidden states of all adapters.

During inference, a critical advantage of Seeded LoRA over other mixture of adapters methods with routers is its ability to merge the adapters into the weights, reducing inference overhead significantly:

$$\mathbf{y} = \mathbf{x}\left(\mathbf{W}_0 + \frac{1}{N}\sum_{i=1}^{N}\mathbf{A}_i\mathbf{B}_i\right) \tag{4}$$

## 4    FINE-TUNING EXPERIMENTS WITH SEEDED LORA

In this section, we evaluate Seeded LoRA compared to LoRA, and MoLoRA fine-tuning. We ensured a similar parameter budget for all models to maintain consistency in our comparison of zero-shot accuracy on various language tasks.

These experiments leverage 9 different datasets[1] for instruction fine-tuning. For details about the dataset composition see Appendix B. The dataset contains a mix of general knowledge, code, and mathematics, totaling 282,360 data points.

In our experiments, we use a multi-stage fine-tuning process to simulate the existence of independent open-source LoRA models. We start with a pretrained LLM, $\mathcal{M}$, trained on a random subset of the training data. We aim to improve $\mathcal{M}$'s performance in $N$ specific areas of expertise. To achieve this, we fine-tune $\mathcal{M}$ with $N$ corresponding datasets, $\mathcal{D} := \{D_1, \ldots, D_N\}$, where each dataset is related to a specific domain. For Seeded LoRA we follow the steps outlined in Section 3, Seeded LoRA. For MoLoRA we finetune on all data – the full data mixture – where the router learns to route to particular expert adapters. For LoRA, we have two experimental settings: (1) finetune on the full data mixture, (2) each LoRA adapter on each individual dataset. In both approaches we control for the overall parameter budget.

| Task Datasets used | LoRA (individual) | LoRA (mixture) | MoLoRA (mixture) | **Seeded LoRA** (mixture) |
|---|---|---|---|---|
| ANLI r1 | 36.10 | **36.30** | 34.80 | 35.20 |
| ANLI r2 | 35.50 | **37.50** | 34.80 | 32.50 |
| ANLI r3 | 34.67 | **38.75** | 32.50 | 34.42 |
| Arc Challenge | 43.77 | 37.20 | 39.59 | **44.45** |
| Arithmetic 2ds | 54.00 | 00.00 | 11.65 | **83.70** |
| Arithmetic 4ds | 37.10 | 00.00 | 14.05 | **52.15** |
| BB Causal Judgement MC | 50.53 | 52.11 | 52.11 | **53.16** |
| Blimp Causative | **76.50** | 74.40 | 75.80 | 75.90 |
| CB | 26.79 | **44.64** | 39.29 | 30.36 |
| COPA | **88.00** | 86.00 | 87.00 | **88.00** |
| HellaSwag | 57.12 | **57.87** | 57.71 | 57.46 |
| RTE | 60.65 | 53.79 | 54.87 | **63.18** |
| TruthfulQA mc1 | 30.35 | 27.29 | 28.64 | **30.35** |
| WIC | 50.16 | **51.25** | 50.94 | 50.00 |
| Winogrande | 69.61 | 70.32 | **71.27** | 70.32 |
| WSC | 39.42 | 37.50 | **45.19** | 38.46 |
| Mean | 50.26 | 44.05 | 45.63 | **52.47** |

Table 1: Zero-shot accuracy of LoRA adapters on individual datasets and the full data mixture for LoRA, MoLoRA, and Seeded LoRA (ours) on multiple evaluation tasks. All models were fine-tuned using instruction-tuning with Llama 2 as the base model. While other full data mixture approaches struggle with task interference on some tasks – particularly Arthimetic 2ds/4ds, RTE, and Arc Challenge – Seeded LoRA shows no signs of task interference. As such, the main benefit of Seeded LoRA can be seen as mitigating interactions between tasks. While individual LoRA finetuning runs do not suffer from task interference, they also do not benefit from data across tasks. Seeded LoRA achieves a good balance of little task interference while still benefiting from data mixtures.

## 4.1 EXPERIMENTAL DETAILS

For training Seeded LoRA experts, we used a rank of 16. The MoLoRA baseline uses the same number of experts as SeededLoRA with the same rank. For the LoRA baseline, we adjusted the rank depending on the number of experts to ensure an equivalent parameter count. All models were fine-tuned using instruction-tuning with Llama 2 7B (Touvron et al., 2023) as the base model.

We assess performance on a range of tasks using the *lm-evaluation-harness* (Gao et al., 2023) and use the task battery used in previous work, MoLoRA (Zadouri et al., 2023). Additionally, to increase the diversity in the task battery and to increase the challenge of multi-task finetuning, we include mathematical arithmetic and reasoning tasks such as Arithmetic 2DS and 4DS (Brown et al., 2020), and the Blimp Causative dataset (Warstadt et al., 2023) in our mixture of tasks.

---

[1]https://huggingface.co/datasets/xxxxx/xxxxx

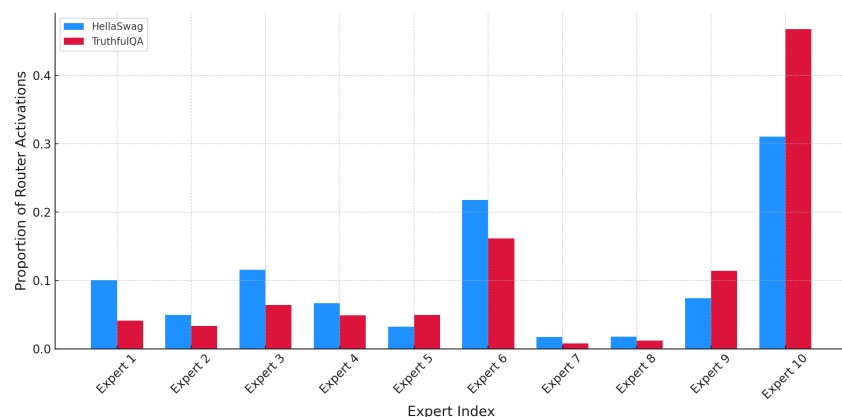

Figure 3: Activation patterns of the routing layer for two distinct tasks, HellaSwag and TruthfulQA, across a set of ten experts. The uniformity in activation distribution suggests similar utilization of experts for both tasks meaning that experts did not specialize in any of the trained 10 domains despite having non-uniform routing probabilities.

These tasks cover diverse areas like natural language inference, arithmetic reasoning, commonsense reasoning, and question answering. Table 1 summarizes the zero-shot accuracy achieved by each model on these tasks. Appendix D contains results for each individual expert in Seeded LoRA.

## 5 RESULTS

As shown in Table 1, Seeded LoRA outperforms LoRA and MoLoRA on average across 16 tasks. Compared to the baseline LoRA and MoLoRA that are trained end-to-end using a multi-task data mixture, Seeded LoRA archives a significant increase of 8.4 and 6.8 points on average accuracy respectively, and outperforms both baselines in 9 tasks. This suggests Seeded LoRA's ability to effectively leverage expert knowledge for broader task applicability.

Notably, Seeded LoRA exhibits superior performance in tasks such as arithmetic reasoning (2D and 4D), where LoRA and MoLoRA struggle. We relate this to task interference and catastrophic forgetting in LoRA and MoLoRA. While LoRA adapters trained on individual datasets do not suffer from task interference, no transfer between tasks takes place. Seeded LoRA strikes a balance between good performance on task mixtures while not suffering from task interference. Overall, Seeded LoRA's performance demonstrates its effectiveness as a fine-tuning approach.

## 6 THE PITFALLS OF ROUTING: ANALYSIS OF SUBTLE ROUTING FAILURES

While the main contribution of this paper is a simple approach that allows for collaborative fine-tuning (CoFT) without any routing, we did extensive experiments of routing approaches. In this section, we highlight subtle failures and show how to debug routing approaches to be able to develop routing methods that might outperform Seeded LoRA.

**Experimental Setup** To investigate the impact of the routing mechanism, we employed Unsupervised Domain Discovery (Gururangan et al., 2023) to cluster a selected dataset into multiple smaller datasets via k-means. Subsequently, a *seed* expert, several domain-specific experts, and a routing layer designed for expert selection through soft merging were developed and trained. Comparative analyses were conducted between Seeded LoRA, LoRA, and MoLoRA, examining configurations with 5, 10, 15, and 30 experts, while ensuring parameter and computational resources remained consistent across these variations.

We then analyzed the routing layer's capability to accurately assign tokens to appropriate experts. To do this we inspect the router probabilities while evaluating with the EleutherAI Eval Harness(Gao

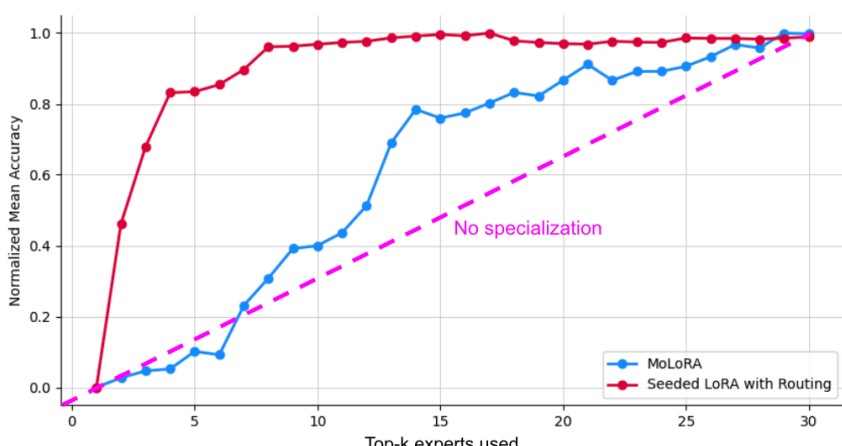

Figure 4: Normalized mean accuracy of Seeded LoRA with Routing, and MoLoRA as a function of the number of top K experts considered for 30 experts. Seeded LoRA, with its independently trained specialized experts, displays a steep increase in performance with a smaller number of top experts, highlighting the benefits of expert specialization. MoLoRA, trained end-to-end, shows a more gradual improvement that is closer to the improvement that would be expected if each additional expert leads to a linear increase in performance.

et al., 2023). We mainly track two quantities: (1) aggregate probability mass over all tokens, (2) normalized proportion of top-k experts activities for all tokens. Normalized proportions means here, we keep track of all top-k expert counts and then divide by the total amount of activated experts.

The evaluation tasks included HellaSwag, testing common-sense reasoning, and TruthfulQA, aimed at addressing failures in truthfulness.

The next paragraphs will discuss subtle routing failures that we observed in these experiments.

**Failure I: Uniform specialization with non-uniform routing** A subtle pattern of expert specialization failure occurs if for a particular evaluation task only a few experts are activated (non-uniform routing probabilities), but that the distribution of experts remains fixed for other tasks. This indicates that all tasks are learned across all experts with one particular weighted average. This pattern is depicted in Figure 3.

**Failure II: Non-uniform routing probability that is unrelated to expert effectiveness** If a router and experts are trained successfully, then adding the top-$k$ experts in order of their routing probability should increase the performance on the end task in a non-linear manner. A non-linear increase indicates, that routing probability $p$ is proportional to the expert effectiveness for that task. A linear increases indicate, while the router assigns a higher $p$ to some experts, all experts are interchangeable and provide similar performance despite different routing probabilities. This is essentially, non-specialization combined with an uncalibrated router that emits non-uniform routing probabilities. See Figure 4 for a failure case that is contrasted with successful specialization.

**Failure III: Uniform routing** While uniform routing where each expert has the same probability $p$ often yields better performance with more experts (Jiang et al., 2024; Muennighoff et al., 2024), we show in Section 7 that this routing pattern is equivalent to grouped convolution and processing. As such, despite its improved performance, uniform routing represents a routing failure since performance of the model is the same with and without routing. Muennighoff et al. (2024) discusses this failure mode as evident when analyzing Mixtral (Jiang et al., 2024).

**Discussion.** Here we depicted common routing failures. Seeded LoRA shows that through the adoption of a *seed* expert and the application of simple averaging of adapter outputs, it is possible to avoid these challenges while simplifying the architecture. We believe that routers can be trained to improve the performance of LoRA MoE approaches, but Seeded LoRA is a strong baseline that we are unable to beat with any current routing approaches (adding routing to Seeded LoRA does

not improve the performance). The failure cases in this section can be used to develop routing mechanisms that improve over Seeded LoRA.

# 7 WHY FAILED ROUTERS CAN STILL BE EFFECTIVE: UNIFORM ROUTING AS GROUPED CONVOLUTIONS

Seeded LoRA uses a simple average of the adapter outputs which is equivalent of a routed architecture where each expert gets the same routing probability. The finding that Seeded LoRA is more effective than architectures that actively route information might be surprising given that routed architectures are often effective in their own right; for example, Mixtral (Jiang et al., 2024) was a very powerful and widely used open-source model at the time of its release, yet it shows the routing failures described above(Muennighoff et al., 2024). In this section, we show that routers that assign equal probability to experts are equivalent to grouped convolution. This highlights that failed routers, while not leading to specialization into experts, might still show improvements in model quality compared to baselines, similarly how grouped convolutional networks such as ResNeXt (Xie et al., 2017) usually outperform regular convolutional networks such as ResNet (He et al., 2015). This also highlights why good model quality alone is not sufficient to determine if a mixture model was trained successfully.

## 7.1 1X1 CONVOLUTION AS MATRIX MULTIPLICATION

Let the input tensor be $X \in \mathbb{R}^{H \times W \times C}$, where:

- $H$ and $W$ represent the spatial dimensions (height and width).
- $C$ is the number of input channels.

Consider a 1x1 convolution kernel denoted by $K \in \mathbb{R}^{1 \times 1 \times C \times F}$, where $F$ corresponds to the number of filters, or output channels. The 1x1 convolution operation can be effectively represented as a matrix multiplication through the following steps:

1. **Reshape the input tensor** $X$: Flatten the spatial dimensions $(H, W)$ into a single dimension, resulting in a matrix $X' \in \mathbb{R}^{HW \times C}$. $X' = \text{reshape}(\mathbf{X}) \in \mathbb{R}^{HW \times C}$.

2. **Reshape the kernel** $K$: Similarly, flatten the spatial dimensions of the kernel and transpose the channel dimensions to obtain a matrix $K' \in \mathbb{R}^{C \times F}$. $K' = \text{reshape}(\mathbf{K}) \in \mathbb{R}^{C \times F}$.

3. **Perform matrix multiplication**: Compute the product of the reshaped input $X'$ and the reshaped kernel $K'$. The resulting matrix $Y' \in \mathbb{R}^{HW \times F}$ represents the flattened form of the convolution's output. Mathematically, this step can be expressed as:$Y' = X'K'$.

4. **Reshape the output**: Finally, reshape the output matrix $Y'$ back to its original tensor format $Y \in \mathbb{R}^{H \times W \times F}$ to obtain the result of the convolution operation.

## 7.2 GROUPED CONVOLUTION AS UNIFORM EXPERT ROUTING

Grouped convolution is an operation where the input tensor $\mathbf{X} \in \mathbb{R}^{H \times W \times C}$ is processed by $k$ independent kernels $\mathbf{K}_i$, and the results are summed: $\mathbf{Y} = \sum \mathbf{X}_i * \mathbf{K}_i$.

When performing uniform routing, this effectively creates a grouped convolution structure with $k$ adapters where the outputs are averaged:$\mathbf{Y} = \frac{1}{k} \sum \mathbf{X} \mathbf{a}_1^i \mathbf{a}_2^i$.

This structure is similar to ResNeXt (Xie et al., 2017), which uses three kernels: reduction, intermediate, and expansion. Our approach simplifies this to two operations (reduction and expansion) while maintaining the grouped computation structure. This allows for efficient computation while still capturing complex transformations through multiple adapter pairs. The only difference between grouped convolution and uniform routing is that grouped convolution uses a sum while uniform routing uses the average of the outputs. This can be rectified with proper initialization.

Specifically, convolutional kernels are often initialized by a normal distribution adjusted for how many channels exist in the kernel (Glorot & Bengio, 2010): $\mathcal{N}\left(0, \sqrt{\frac{1}{C}}\right)$. Here $C$ represents the channel dimension.

For uniform routing to be equivalent in the output distribution, we can initialize each LoRA adapter with $\mathcal{N}\left(0, \sqrt{\frac{1}{C \times k}}\right)$. Here, $C$ represents the input dimension, and $k$ is the number of expert adapters used.

With this specific initialization scheme, the variance of the Seeded LoRA output will align with that of the grouped convolutions, leading to equivalent behavior.

## 8 RELATED WORK

**Mixture-of-Adapters (MoA) methods**   MoLoRA (Zadouri et al., 2023) integrating Low-Rank Adapters as experts, updates only a small portion of parameters, efficiently enhancing performance across various tasks. Similarly, SiRA (Zhu et al., 2023) adopts a Sparse MoE strategy, implementing a top-k expert routing with limits on token processing and an expert dropout to combat overfitting, aiming for computational efficiency. PHATGOOSE (Muqeeth et al., 2024) facilitates zero-shot generalization by routing among language model experts and employing a sigmoid gate for efficient top-k inference routing. Lastly, LoRAMoE (Dou et al., 2024) safeguards world knowledge within LLMs during fine-tuning by freezing the main model and fine-tuning select LoRAs, thus bolstering downstream task performance while preserving the original knowledge base.

**Branch-Train-Merge (BTM) Approaches**   BTM (Li et al., 2022) is a communication-efficient algorithm designed for the parallel training of large language models (LLMs). It facilitates the independent training of model subparts, termed Expert Language Models (ELMs), across different data subsets. ELMs form the ELMFOREST and can be dynamically modulated or integrated through ensembling or parameter averaging. Cluster-Branch-Train-Merge (c-BTM) (Gururangan et al., 2023) extends BTM by incorporating unsupervised domain discovery, enabling domain-specific training and forming a sparse ensemble for efficient inference. Branch-Train-MiX (Sukhbaatar et al., 2024) further advances this paradigm by mixing trained domain-specific experts into an MoE model, yielding an efficient LLM with enhanced accuracy-efficiency trade-offs.

## 9 LIMITATIONS & FUTURE WORK

Despite Seeded LoRA's demonstrated efficacy in enhancing the zero-shot performance of LLMs across a variety of tasks while adding chatbot capabilities to pretrained models, there are inherent limitations that require further exploration:

**Past models are unseeded.** While future models can be initialized via Seeded LoRA currently available models are not seeded and as such not initialized in the same optimization subspace.

**Inherent limits of averaging.** While a simple average of expert outputs works in Seeded LoRA, this has inherent limits as ineffective experts add more and more noise decreasing the signal to noise ratio. As such, when too many experts are merged more advanced weighted averaging techniques will become necessary.

**Scalable Expert Management.** To address scalability, further research could look on developing efficient algorithms for expert selection and routing that minimize computational overhead. Techniques such as sparse expert selection, where only a subset of the most relevant experts are activated for a given input, could improve Seeded LoRA's performance.

**Number of Experts.** Determining an optimal number of experts for a given task or dataset remains an open question. Techniques for dynamically adjusting the number of experts based on task complexity or data characteristics could be beneficial.

REPRODUCIBILITY STATEMENT

The code to fine-tune LoRA, MoLoRa, and Seeded LoRA, as well as the evaluation code, can be found in Github[2].

---

[2]https://github.com/xxxxx/xxxxx

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

## A CONVOLUTIONS

**Convolutional Neural Networks (CNNs)**    CNNs automate feature extraction from images using layers of convolutional kernels. These kernels, through the convolution operation, identify patterns and features within the input data, making them essential for tasks such as image and video recognition, image classification, and medical image analysis. The convolution operation is mathematically represented as:

$$F(i, j) = (K * X)(i, j) = \sum_m \sum_n K(m, n) X(i - m, j - n) \tag{5}$$

where $F$ is the feature map resulting from applying the kernel $K$ to the input image $X$ at coordinates $(i, j)$.

**Convolutional Kernels**    Convolutional kernels are the core components of CNNs, allowing the network to capture spatial hierarchies of features. Early layers might capture basic patterns such as edges and textures, while deeper layers combine these features to detect more complex patterns. The design of CNN architectures such as ResNet (He et al., 2015) demonstrates how deep networks can effectively learn a wide variety of features by applying convolutional kernels across multiple layers.

**Grouped Convolutions**    Grouped convolutions, introduced in  (Krizhevsky et al., 2012), extend the convolutional operation by dividing the input and kernels into groups, allowing each group to perform convolutions independently. This method reduces computational requirements and parameters while maintaining the network's effectiveness. ResNeXt (Xie et al., 2017) leverages grouped convolutions, introducing the concept of cardinality to efficiently scale the model's capacity. This approach demonstrates the significant advantages of grouped convolutions in deep learning architectures:

$$F_g = K_g * X_g \tag{6}$$

where $F_g$ represents the feature map produced by the $g^{th}$ group's convolution of kernel $K_g$ with input $X_g$.

## B DATA DETAILS

The experiments for Seeded LoRA leverage a composite dataset for instruction fine-tuning containing a mix of general knowledge, code, and mathematics, totaling 282,360 data points. Instruction fine-tuning contrasts with traditional supervised fine-tuning, which primarily aims to correlate input data with corresponding outputs. The data originates from various sources:

- `Open-Orca/OpenOrca` (Lian et al., 2023): collection of augmented FLAN (Wei et al., 2022) data that aligns with the distributions outlined in the Orca paper (Mukherjee et al., 2023).
- `TokenBender/code_instructions_122k_alpaca_style`[3]:  coding questions following the Alpaca template.
- `camel-ai/math` (Li et al., 2023): composed of 50K problem-solution pairs obtained using GPT-4.
- `yahma/alpaca-cleaned` (Taori et al., 2023): contains a cleaned version of the Alpaca dataset.
- `garage-bAInd/Open-Platypus` (Lee et al., 2023): dataset focused on improving LLM logical reasoning skills and was used to train the Platypus2 models.
- `sahil2801/CodeAlpaca-20k` (Chaudhary, 2023): contains 20K code problems in the Alpaca format.

---

[3]https://huggingface.co/datasets/TokenBender/code_instructions_122k_alpaca_style

- `c-s-ale/dolly-15k-instruction-alpaca-format`: cleaned and alpaca formatted version of Dolly (Conover et al., 2023), a corpus of more than 15,000 records generated by thousands of Databricks employees.

- `hendrycks/competition_math` (Hendrycks et al., 2021): consists of problems from mathematics competitions, including the AMC 10, AMC 12, AIME, and more. Each problem has a full step-by-step solution, which can be used to teach models to generate answer derivations and explanations.

- `gsm8k` (Cobbe et al., 2021): dataset of 8.5K high quality linguistically diverse grade school math word problems. The dataset was created to support the task of question answering on basic mathematical problems that require multi-step reasoning.

| Dataset | Count | Percentage (%) |
|---|---|---|
| `Open-Orca/OpenOrca` | 70565 | 24.99 |
| `TokenBender/code_instructions_122k_alpaca_style` | 60979 | 21.60 |
| `camel-ai/math` | 50000 | 17.71 |
| `yahma/alpaca-cleaned` | 25880 | 9.17 |
| `garage-bAInd/Open-Platypus` | 24926 | 8.83 |
| `sahil2801/CodeAlpaca-20k` | 20022 | 7.09 |
| `c-s-ale/dolly-15k-instruction-alpaca-format` | 15015 | 5.32 |
| `hendrycks/competition_math` | 7500 | 2.66 |
| `gsm8k` | 7473 | 2.65 |

Table 2: Distribution of elements and their respective percentages across various datasets.

For training Seeded LoRA experts, the following hyperparameters were used:

- Rank: 16 (dimensionality of the low-rank adaptation space)
- LoRA Alpha: 8
- LoRA Dropout: 0.05
- Epochs: 2

## C    SEEDED LORA FINE-TUNING USING UNSUPERVISED DOMAIN DISCOVERY

Building upon the foundation laid by c-BTM (Gururangan et al., 2023), we experimented with Unsupervised Domain Discovery to create clusters to train experts.

The process begins by segmenting the data using k-means clustering. This divides the data (denoted by $X$ with $N$ samples) into $K$ distinct clusters ($C$). Each cluster is characterized by a centroid ( $\mu_j$ ), representing the average feature vector of its members. The k-means algorithm aims to minimize the inertia, ensuring data points within each cluster are similar.

$$\sum_{i=0}^{n} \min_{\mu_j \in C} (\|x_i - \mu_j\|^2)$$

This method is flexible in its data representation. You can use any encoding method that captures the dataset's information suitable for unsupervised domain discovery. In this case, we create embeddings of our data. Afterwards, experts are trained as shown in Section 4.

Results for this setting can be seen in Appendices G and H.

## D    EVALUATION RESULTS FOR SEEDED LORA EXPERTS TRAINED ON INDIVIDUAL DATASETS

Table 3 contains the evaluation resutls for Seeded LoRA with experts trained on the individual datasets.

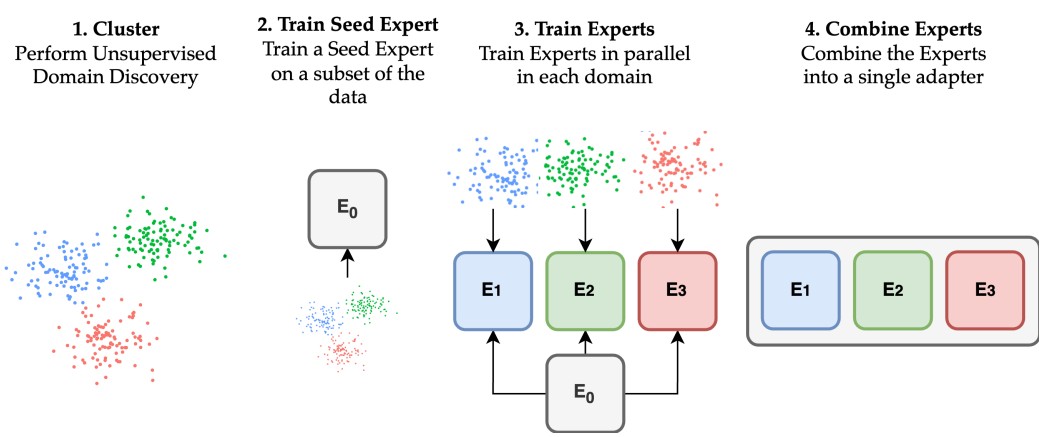

Figure 5: Seeded LoRA fine-funing using Unsupervised Domain Discovery. This method contains four stages: 1) **Unsupervised Domain Discovery** on an unlabelled dataset to create domain-specific datasets. 2) *Seed Expert* **Training**. 3) *Cluster-Specific Expert* **Training**. 4) **Expert Combination**.

| Task | Seed Expert | Exp. 0 | Exp. 1 | Exp. 2 | Exp. 3 | Exp. 4 | Exp. 5 | Exp. 6 | Exp. 7 | Exp. 8 |
|---|---|---|---|---|---|---|---|---|---|---|
| ANLI r1 | 36.10 | 34.70 | 35.30 | 33.90 | 34.00 | 31.90 | 35.20 | 37.00 | 34.70 | 38.00 |
| ANLI r2 | 35.50 | 34.20 | 34.30 | 33.40 | 33.40 | 31.20 | 34.30 | 37.50 | 32.30 | 33.90 |
| ANLI r3 | 34.67 | 35.92 | 34.33 | 32.58 | 33.42 | 32.67 | 33.50 | 35.83 | 33.25 | 35.33 |
| Arc Challenge | 43.77 | 34.73 | 46.16 | 40.96 | 45.56 | 42.32 | 45.22 | 44.54 | 42.75 | 43.34 |
| Arithmetic 2ds | 54.00 | 00.00 | 94.50 | 46.35 | 72.65 | 58.45 | 91.10 | 96.25 | 53.30 | 43.65 |
| Arithmetic 4ds | 37.10 | 00.00 | 52.80 | 39.45 | 44.70 | 39.90 | 50.35 | 58.35 | 37.90 | 37.00 |
| BB C.J. MC | 50.53 | 52.11 | 47.89 | 49.47 | 47.89 | 54.21 | 50.53 | 52.63 | 52.11 | 51.58 |
| Blimp Causative | 76.50 | 68.10 | 75.00 | 75.30 | 74.10 | 73.70 | 76.40 | 75.70 | 77.80 | 77.30 |
| CB | 26.79 | 26.79 | 39.29 | 21.43 | 30.36 | 21.43 | 28.57 | 32.14 | 28.57 | 26.79 |
| COPA | 88.00 | 88.00 | 86.00 | 85.00 | 88.00 | 86.00 | 87.00 | 88.00 | 88.00 | 88.00 |
| HellaSwag | 57.12 | 57.97 | 57.04 | 56.84 | 57.77 | 57.25 | 57.12 | 57.42 | 57.38 | 57.22 |
| RTE | 60.65 | 52.71 | 63.18 | 51.62 | 64.98 | 58.84 | 61.01 | 59.21 | 59.93 | 58.84 |
| TruthfulQA mc1 | 30.35 | 31.95 | 28.52 | 33.90 | 34.39 | 31.09 | 27.91 | 28.76 | 28.89 | 28.03 |
| WIC | 50.16 | 50.00 | 49.69 | 50.00 | 49.84 | 50.00 | 50.00 | 50.00 | 50.00 | 50.00 |
| Winogrande | 69.61 | 70.96 | 71.35 | 68.51 | 70.72 | 70.24 | 70.40 | 69.77 | 69.69 | 69.77 |
| WSC | 39.42 | 36.54 | 40.38 | 36.54 | 41.35 | 36.54 | 50.96 | 37.50 | 41.35 | 38.46 |
| Average | 50.26 | 42.16 | 53.48 | 47.20 | 51.44 | 48.48 | 53.09 | 53.78 | 49.24 | 48.57 |

Table 3: Zero-shot accuracy of Seeded LoRA experts on multiple evaluation tasks for experts trained on individual datasets.

# E EVALUATION RESULTS FOR SEEDED LORA TRAINED ON INDIVIDUAL DATASETS WITH NO *Seed Expert*

Table 4 contains the evaluation resutls for Seeded LoRA trained on individual datasets with no *Seed Expert*.

# F SEEDED LORA WITH SEED EXPERT IN THE FINAL MODEL

We also experimented with including the *Seed Expert* in the final model. In Seeded, this resulted in an average score of 52.29. This shows that the general knowledge acquired by the *seed* expert is not lost when the rest of the experts are trained.

# G EVALUATION RESULTS FOR SEEDED LORA EXPERTS TRAINED ON CLUSTERS

Table 6 contains the results for Seeded LoRA trained on clusters using Unsupervised Domain Discovery. Table 7 contains the results for each expert in this model.

| Task | Seeded LoRA (No Seed) | Exp. 0 | Exp. 1 | Exp. 2 | Exp. 3 | Exp. 4 | Exp. 5 | Exp. 6 | Exp. 7 | Exp. 8 |
|------|------|------|------|------|------|------|------|------|------|------|
| ANLI r1 | 37.10 | 36.40 | 37.00 | 33.20 | 38.20 | 32.20 | 35.60 | 35.70 | 36.30 | 35.90 |
| ANLI r2 | 38.60 | 34.80 | 38.30 | 31.50 | 37.80 | 31.60 | 37.10 | 38.60 | 37.60 | 36.80 |
| ANLI r3 | 37.83 | 35.25 | 35.42 | 33.25 | 36.25 | 32.00 | 37.67 | 36.08 | 37.17 | 37.83 |
| Arc Challenge | 43.34 | 34.22 | 44.80 | 40.36 | 44.54 | 42.24 | 43.43 | 44.28 | 42.58 | 42.66 |
| Arithmetic 2ds | 49.80 | 00.00 | 93.90 | 46.00 | 44.55 | 44.45 | 51.90 | 52.85 | 50.10 | 48.90 |
| Arithmetic 4ds | 37.85 | 00.00 | 45.15 | 40.20 | 41.90 | 38.40 | 36.25 | 35.80 | 37.05 | 36.70 |
| BB C.J. MC | 52.63 | 51.05 | 47.37 | 48.42 | 48.95 | 51.05 | 55.79 | 53.16 | 51.58 | 52.63 |
| Blimp Causative | 74.00 | 63.70 | 74.10 | 75.30 | 73.70 | 75.00 | 75.30 | 73.20 | 74.00 | 73.90 |
| CB | 39.29 | 26.79 | 37.50 | 30.36 | 28.57 | 17.86 | 48.21 | 44.64 | 48.21 | 53.57 |
| COPA | 87.00 | 90.00 | 87.00 | 84.00 | 88.00 | 87.00 | 87.00 | 87.00 | 88.00 | 87.00 |
| HellaSwag | 57.38 | 57.81 | 57.14 | 56.82 | 57.35 | 57.28 | 57.14 | 57.41 | 57.26 | 57.07 |
| RTE | 62.82 | 52.71 | 59.21 | 54.51 | 63.18 | 59.57 | 61.37 | 57.04 | 62.45 | 62.45 |
| TruthfulQA mc1 | 28.64 | 31.95 | 27.29 | 33.05 | 33.05 | 30.35 | 25.21 | 27.42 | 25.46 | 24.36 |
| WIC | 50.00 | 50.00 | 50.47 | 50.00 | 50.00 | 50.00 | 50.00 | 50.00 | 49.53 | 49.84 |
| Winogrande | 70.32 | 71.59 | 70.09 | 68.90 | 70.01 | 70.56 | 69.69 | 69.30 | 69.06 | 69.14 |
| WSC | 38.46 | 36.54 | 44.23 | 36.54 | 36.54 | 36.54 | 38.46 | 37.50 | 38.46 | 38.46 |
| Average | 50.31 | 42.04 | 53.05 | 47.65 | 49.53 | 47.25 | 50.63 | 49.99 | 50.30 | 50.45 |

Table 4: Zero-shot accuracy of Seeded, trained on individual datasets without Seed Expert, on multiple evaluation tasks.

| Task | Seeded LoRA with *seed* expert |
|------|------|
| ANLI r1 | 35.70 |
| ANLI r2 | 32.90 |
| ANLI r3 | 34.75 |
| Arc Challenge | 44.71 |
| Arithmetic 2ds | 81.85 |
| Arithmetic 4ds | 49.15 |
| BB Causal Judgement MC | 48.95 |
| Blimp Causative | 77.20 |
| CB | 30.36 |
| COPA | 88.00 |
| HellaSwag | 57.39 |
| RTE | 64.98 |
| TruthfulQA mc1 | 31.82 |
| WIC | 50.00 |
| Winogrande | 70.48 |
| WSC | 38.46 |
| Average | 52.29 |

Table 5: Zero-shot accuracy of Seeded LoRA with *seed* expert in the final model trained on clusters on multiple evaluation tasks.

# H EVALUATION RESULTS FOR SEEDED LoRA TRAINED ON CLUSTERS WITH NO *Seed Expert*

Table 8 contains the evaluation results for Seeded LoRA trained on clusters with no *Seed Expert*.

| Task | Seeded LoRA |
|---|---|
| ANLI r1 | 34.90 |
| ANLI r2 | 32.10 |
| ANLI r3 | 35.08 |
| Arc Challenge | 44.62 |
| Arithmetic 2ds | 85.20 |
| Arithmetic 4ds | 50.25 |
| BB Causal Judgement MC | 50.00 |
| Blimp Causative | 77.30 |
| CB | 26.79 |
| COPA | 88.00 |
| HellaSwag | 57.48 |
| RTE | 64.98 |
| TruthfulQA mc1 | 31.46 |
| WIC | 50.00 |
| Winogrande | 70.48 |
| WSC | 37.50 |
| Average | 52.25 |

Table 6: Zero-shot accuracy of Seeded LoRA trained on clusters on multiple evaluation tasks.

| Task | Seed Expert | Exp. 0 | Exp. 1 | Exp. 2 | Exp. 3 | Exp. 4 | Exp. 5 | Exp. 6 | Exp. 7 | Exp. 8 |
|---|---|---|---|---|---|---|---|---|---|---|
| ANLI r1 | 36.10 | 33.20 | 36.30 | 37.10 | 34.00 | 34.10 | 33.50 | 34.20 | 36.50 | 36.40 |
| ANLI r2 | 35.50 | 32.70 | 33.80 | 34.10 | 32.80 | 33.40 | 31.30 | 31.90 | 32.70 | 32.50 |
| ANLI r3 | 34.67 | 33.25 | 35.92 | 33.58 | 32.67 | 32.92 | 33.08 | 33.33 | 33.33 | 32.50 |
| Arc Challenge | 43.77 | 41.89 | 35.92 | 46.08 | 43.94 | 42.83 | 41.81 | 43.69 | 45.05 | 45.22 |
| Arithmetic 2ds | 54.00 | 83.70 | 00.00 | 92.30 | 45.95 | 93.25 | 32.85 | 76.35 | 67.90 | 79.20 |
| Arithmetic 4ds | 37.10 | 49.35 | 00.00 | 45.45 | 38.50 | 55.35 | 33.60 | 43.10 | 41.70 | 44.35 |
| BB C.J. MC | 50.53 | 52.11 | 52.11 | 47.89 | 47.37 | 51.58 | 48.95 | 50.53 | 47.89 | 47.37 |
| Blimp Causative | 76.50 | 73.00 | 65.60 | 75.80 | 77.60 | 76.90 | 78.00 | 77.50 | 76.00 | 75.40 |
| CB | 26.79 | 44.64 | 35.71 | 19.64 | 16.07 | 26.79 | 23.21 | 35.71 | 26.79 | 26.79 |
| COPA | 88.00 | 87.00 | 90.00 | 88.00 | 87.00 | 88.00 | 86.00 | 88.00 | 87.00 | 88.00 |
| HellaSwag | 57.12 | 57.47 | 57.93 | 57.48 | 57.24 | 57.31 | 56.94 | 57.03 | 57.17 | 57.33 |
| RTE | 60.65 | 51.62 | 53.07 | 62.82 | 65.70 | 63.18 | 64.26 | 59.57 | 61.73 | 63.18 |
| TruthfulQA mc1 | 30.35 | 30.60 | 32.19 | 30.48 | 31.33 | 32.19 | 33.05 | 31.21 | 30.72 | 30.11 |
| WIC | 50.16 | 50.00 | 50.00 | 50.00 | 49.53 | 50.00 | 50.00 | 50.00 | 50.31 | 50.16 |
| Winogrande | 69.61 | 70.88 | 70.64 | 71.03 | 70.48 | 71.51 | 70.48 | 71.82 | 70.48 | 70.48 |
| WSC | 39.42 | 36.54 | 36.54 | 41.35 | 52.88 | 36.54 | 36.54 | 36.54 | 47.12 | 41.35 |
| Average | 49.39 | 51.74 | 42.85 | 52.06 | 48.94 | 52.86 | 47.09 | 51.28 | 50.77 | 51.27 |

Table 7: Zero-shot accuracy of Seeded LoRA experts, each one trained on a different cluster, on multiple evaluation tasks.

| Task | Seeded LoRA (No Seed) | Exp. 0 | Exp. 1 | Exp. 2 | Exp. 3 | Exp. 4 | Exp. 5 | Exp. 6 | Exp. 7 | Exp. 8 |
|---|---|---|---|---|---|---|---|---|---|---|
| ANLI r1 | 38.60 | 33.80 | 34.90 | 37.30 | 37.10 | 36.40 | 37.50 | 37.50 | 36.10 | 36.40 |
| ANLI r2 | 37.30 | 32.20 | 34.20 | 38.90 | 38.30 | 34.10 | 36.90 | 37.10 | 37.90 | 36.50 |
| ANLI r3 | 37.00 | 33.75 | 32.50 | 38.25 | 36.92 | 35.42 | 37.83 | 37.75 | 38.17 | 37.83 |
| Arc Challenge | 43.52 | 43.77 | 42.75 | 43.94 | 43.94 | 42.92 | 43.26 | 43.09 | 42.83 | 43.69 |
| Arithmetic 2ds | 51.75 | 75.60 | 89.70 | 65.75 | 49.80 | 47.50 | 49.95 | 49.40 | 50.40 | 50.30 |
| Arithmetic 4ds | 37.45 | 51.85 | 54.45 | 36.75 | 36.75 | 38.00 | 36.60 | 36.85 | 36.35 | 36.25 |
| BB C.J. MC | 53.68 | 52.11 | 52.63 | 50.00 | 48.95 | 48.42 | 48.42 | 46.32 | 51.05 | 52.11 |
| Blimp Causative | 74.70 | 75.70 | 71.00 | 75.10 | 73.80 | 77.30 | 74.30 | 77.20 | 73.60 | 74.10 |
| CB | 39.29 | 33.93 | 08.93 | 32.14 | 44.64 | 28.57 | 44.64 | 39.29 | 42.86 | 42.86 |
| COPA | 88.00 | 86.00 | 88.00 | 88.00 | 87.00 | 87.00 | 87.00 | 86.00 | 87.00 | 88.00 |
| HellaSwag | 57.20 | 57.32 | 57.60 | 57.22 | 57.09 | 57.06 | 56.87 | 57.08 | 57.28 | 57.14 |
| RTE | 63.54 | 66.06 | 62.82 | 60.29 | 63.54 | 62.82 | 62.45 | 63.18 | 63.54 | 62.82 |
| TruthfulQA mc1 | 28.52 | 29.38 | 34.39 | 27.91 | 28.27 | 30.35 | 28.27 | 29.74 | 24.97 | 25.58 |
| WIC | 50.00 | 50.31 | 50.00 | 50.00 | 49.53 | 50.16 | 49.69 | 49.69 | 49.84 | 49.84 |
| Winogrande | 69.38 | 70.32 | 70.88 | 69.77 | 69.53 | 70.01 | 69.22 | 69.61 | 68.90 | 69.61 |
| WSC | 38.46 | 36.54 | 36.54 | 38.46 | 39.42 | 45.19 | 38.46 | 40.38 | 37.50 | 38.46 |
| Average | 50.52 | 51.79 | 51.33 | 50.61 | 50.28 | 49.45 | 50.08 | 50.01 | 49.89 | 50.09 |

Table 8: Zero-shot accuracy of Seeded LoRA, trained on Clusters without Seed Expert, on multiple evaluation tasks.

