# OpenReview forum: "Seeded LoRA: Collaborative Fine-Tuning Through Seed Initialization of Adapters"
_ICLR.cc/2025/Conference — Submitted to ICLR 2025_

### Official Review · Reviewer_c3Qw · 2024-10-24

**Soundness:** 3
**Presentation:** 3
**Contribution:** 2
**Rating:** 5
**Confidence:** 4

**Summary:**

This paper introduces Seeded LoRA, a novel Collaborative Fine-Tuning method that eliminates the need for post-merge fine-tuning of large language models. Specifically, Seeded LoRA generates a generic seed expert LoRA that is fine-tuned on a small random subset of the dataset, which is then used to initialize other downstream task LoRAs. Extensive experiments on natural language processing benchmarks demonstrate the effectiveness of Seed LoRA.

**Strengths:**

The idea of the proposed Seeded LoRA seems to be simple and effective. The analysis of routing failures in MoLoRA is also impressive. The paper is well-written and easy to follow.

**Weaknesses:**

1. The paper should discuss how CoFT relates to and differs from multi-task/continuous learning, which also aims for improved model performance. A summarizing table could emphasize the necessity of the CoFT task.

2. In Dataset-Specific Expert fine-tuning, does the seeded LoRA recognize the dataset during training (thereby updating domain-specific LoRA)? If so, it may introduces extra information about data distributions, making comparison unfair in Table 1.

3. The authors' approach of using the same subspace for initializing task-specific LoRAs opposes O-LoRA's method of using orthogonal subspaces and merging them after continued learning. More discussion and comparison with O-LoRA are needed.
[1] EMNLP24 Orthogonal Subspace Learning for Language Model Continual Learning

**Questions:**

Please see the weakness section.

---

> ### Author Response · Authors · 2024-11-19
>
> Thank you so much for taking the time to review or work and suggesting improvements. We appreciate it and will try to incorporate as many of them as possible in our final submission. Below, we add responses to some of the questions raised. We hope this additional information provides a complete picture of our work and contributions.
>
> **Q1: discuss how CoFT relates to and differs from multi-task/continuous learning, which also aims for improved model performance**
>
> A1: CoFT is an interesting middle-point between federated learning and multi-task learning. In multi-task learning one often has access to all the data. In the case of CoFT, you only have the weights of existing models and a single new task/dataset. The inherent difficulty is to do multi-task learning and (1) you only have one dataset + weight, (2) you do not know what the weights have been trained on exactly. The difference to continuous learning is that CoFT is that continuous learning gradually adapts, while in CoFT the main artifacts that are shared are trained models after certain checkpoints. We will add clarification and more context comparing CoFT/our method with continual/multi-task learning approaches.
>
> **Q2: In Dataset-Specific Expert fine-tuning, does the seeded LoRA recognize the dataset during training (thereby updating domain-specific LoRA)?**
>
> A2: It does not need to recognize the dataset because all LoRA adapters are trained independently. So (1) a seed expert is trained on a specific dataset, (2) at test time all LoRA adapters are merged (including the new dataset-specific expert), (3) this approach yields high quality domain-specific performance without knowing the input domain (no routing required). This stands in contrast to almost all routing work, where the incoming data needs to be recognized for domain-specialization.
>
> **Q3: The authors' approach of using the same subspace for initializing task-specific LoRAs opposes O-LoRA's method of using orthogonal subspaces and merging them after continued learning. More discussion and comparison with O-LoRA are needed.**
>
> A3: Geometrically, one can have merging without interference if (1) LoRAs are orthogonal, (2) LoRA start from the same weight space and only are optimized sparsely. O-LoRA does (1), Seeded LoRA does (2): Once the optimization landscape has settled (seed training; picture going down an optimization canyon), then most weights do not move anymore while only some move heavily. Depending on the data used, different dimensions move, while most dimensions remain steady. One does not have this property without a seed LoRA, since at the beginning of LoRA finetuning all weights move until they have settled into the local optimization landscape.

---

> ### Comment · Reviewer_c3Qw · 2024-11-20
>
> Thank you for your clarification.
>
> Regarding Q2, a clearer process is needed.
>
> 1. When you said, "all LoRA adapters are trained independently," does it mean that after a seed expert is initialized, for each dataset, a new LoRA model initialized with that seed expert is trained? For instance, for 10 datasets, there are 10 LoRAs trained independently?
>
> 2. For the comparison methods, including lora-mixture and MoLoRA-mixture, does it mean that the datasets are merged and fine-tuned on a multi-task dataset?

---

> > ### Author Response · Authors · 2024-11-20
> >
> > Hello,
> >
> > > When you said, "all LoRA adapters are trained independently," does it mean that after a seed expert is initialized, for each dataset, a new LoRA model initialized with that seed expert is trained? For instance, for 10 datasets, there are 10 LoRAs trained independently?
> >
> > Yes, that is exactly how it works.
> >
> > > For the comparison methods, including lora-mixture and MoLoRA-mixture, does it mean that the datasets are merged and fine-tuned on a multi-task dataset?
> >
> > Yes, that is correct.
> >
> >
> > We will clarify this in the final submission to make it clear.

---

> ### Author Response · Authors · 2024-11-21
>
> As the discussion period is nearing its end, we wanted to ask Reviewer c3Qw if there are any follow-up points we can clarify. We have responded to all concerns raised. If there are no further points of clarification regarding the manuscript, experiments, our fine-tuning procedure, and details about the optimization subspace that makes our method possible, we would kindly ask that reviewer c3Qw to consider increasing their score to reflect this.

---

> > ### Comment · Reviewer_c3Qw · 2024-11-22
> >
> > Thank you for your reply.
> >
> > As I noted in Q2, your method uses the attribute information of the dataset to determine which samples belong to which dataset, making the comparison inherently unfair. This is particularly for the MoLoRA, which does not rely on the dataset's attribute information. Although the authors have conducted supplementary experiments omitting attribute information, i.e., unsupervised domain discovery, it remains possible that MoLoRA could outperform the proposed method if such additional information were incorporated. Therefore, a more comprehensive comparison is necessary to make results more convincing.
> >
> > Additionally, the comparison and analysis with O-LoRA is also important, as it can serve as a comparison for different initialization methods for model merging.

---

### Official Review · Reviewer_ouui · 2024-10-29

**Soundness:** 3
**Presentation:** 2
**Contribution:** 2
**Rating:** 5
**Confidence:** 3

**Summary:**

This paper introduces a novel method for collaborative fine-tuning that eliminates the need for costly post-merging fine-tuning, typically required to enhance the accuracy of merged models. The authors contribute by demonstrating that initializing a model within the same optimization subspace ensures that domain- or task-specific fine-tuning preserves this shared subspace. Consequently, the final model can be obtained as a straightforward average of experts, rather than relying on complex routing mechanisms, as seen in mixture-of-experts (MoE) models. The authors named their method Seeded LoRA.

The authors propose seeding the initial model using a random subset of datasets from multiple tasks, then creating a mixture of experts using their proposed approach. Experimental results demonstrate that this method surpasses alternative techniques, including LoRA and MoLoRA.

**Strengths:**

Strengths of the paper:

1. *Conceptual Foundation of the Approach*: **Seeded LoRA** is based on the concept of ensuring a shared optimization subspace across different tasks. This alignment allows the weights of the fine-tuned models to be linearly combined, resulting in a model resembling a mixture-of-experts (MoE) without complex routing mechanisms.

2. The Proposed Framework is **simple** to understand and **intuitive**.

3. Experiments showing the efficacy of the proposed approach are shown on LLaMa 2, using a variety of tasks including inference, reasoning, and QA.

4. Limitations and applications of the proposed method are well documented.

**Weaknesses:**

These are some Weaknesses in my opinion:

1. Results are currently presented on only one model, and that is an older version of LLaMa. It would strengthen the paper to include evaluations on newer models, such as LLaMa 3, Vicuna, or Bloom, or even on smaller models like RoBERTa with fine-tuning instead of instruction tuning. Testing across 1-2 additional open-source models that support fine-tuning or instruction tuning could provide a broader and more relevant context for the findings.

2. Unfair comparisons, especially LoRA(mixture). Please check the question below regarding this weakness.

3. Limited ablation studies. Impact of the percentage(5% vs 10%, mentioned in the paper but no experiments) of seed dataset, different datasets used in seed dataset(using a subset of dataset not data samples for seeding), biased sampling(sampling more from one dataset than other), etc etc. There are a bunch of parameters that can affect the performance of the proposed method which are not shown. Refer to Question 2 for more details of this weakness.

**Questions:**

These are some of the areas that weren't very clear to me or results didn't make sense:

1. When fine-tuning with LoRA (mixture), are you using a single adapter, or does the number of adapters match LoRA (individual)? Explaining in more detail would clarify the doubt.

Based on the results, I guess that it is a single adapter is used for the entire LoRA (mixture) dataset, with an increased rank. However, this may not provide a fair comparison, as the diverse gradients from different tasks could impact model performance, with the sampling strategy and dataset ratio playing significant roles in final accuracy. This might explain why the model shows zero accuracy on mathematical tasks. Instead, I suggest a variant of LoRA (individual) in which all LoRAs are applied simultaneously during fine-tuning on the mixture dataset. Would this approach allow each LoRA to automatically specialize for distinct tasks during convergence? even if it doesn't, I think this would be a fair comparison.

2. While I agree that initializing or seeding the model with a dataset can yield better initializations, isn’t a trained model already positioned within an optimized subspace? How, then, would additional seeding impact performance? Furthermore, wouldn't a new seed be necessary whenever a new dataset or task is added to the mixture?

(a) A crucial ablation study would be to evaluate performance on out-of-distribution datasets. Here, I’m referring to datasets outside the current mixture but within the same task scope, such as other inferences or QA sets. This would clarify whether re-seeding is essential for a new dataset of the same task.

(b) For a completely new task, however, I would certainly expect that re-seeding would be required, but proving that with an ablation would make the experiment section stronger. Assess the impact of adding new datasets or tasks to the mixture without re-seeding. These experiments would provide valuable insights into the robustness and generalizability of the Seeded LoRA approach.



3.I am uncertain about the purpose of Section 7, as it seems to add little beyond what was already covered in Seeded LoRA. The section does not enhance my understanding of why Seeded LoRA offers a distinct advantage. Moving this content to the Appendix may be more appropriate, as the findings appear to be either known or intuitive enough to understand without a dedicated section.

---

> ### Author Response · Authors · 2024-11-19
>
> Thank you so much for taking the time to review or work and suggesting improvements. We appreciate it and will try to incorporate as many of them as possible in our final submission. Below, we add responses to some of the questions raised. We hope this additional information provides a complete picture of our work and contributions.
>
> **Q1: Testing across 1-2 additional open-source models that support fine-tuning or instruction tuning could provide a broader and more relevant context for the findings.**
>
> A1:We acknowledge that more experiments could add to the robustness of our results, but we lack the computational resources to complete more training runs. Note that we carefully control for data and different methods for many different datasets/setups. Overall, this gives us a large sample size which makes our results already quite robust. We run experiments on a setup that extends the experiments in the MoLoRA work, which was accepted to ICLR2024, so our experiments have a larger diversity than previously established work. The main result is that for the first time we are able to merge models without losing significant performance if the target tasks are diverse.
>
> **Q2: When fine-tuning with LoRA (mixture), are you using a single adapter, or does the number of adapters match LoRA (individual)?**
>
> A2: For LoRA (mixture) we use a single adapter trained on all data with a parameter budget equivalent to the one used in the other methods. In this case, we’re comparing performance across tasks for a fixed parameter budget, and show that LoRA (mixture) as well as MoLoRA, suffer from task inference.
>
> **Q3: I suggest a variant of LoRA (individual) in which all LoRAs are applied simultaneously during fine-tuning on the mixture dataset.**
>
> A3: Thanks for the suggestion, we will update table 1 to better showcase results. The difficulty of combining multiple LoRA adapters that are not seeded is what motivated this work in the first place. We experimented with applying all LoRAs simultaneously during fine-tuning. This setting is equivalent to SeededLoRA without a seed expert, and results for it can be found in Appendix H, table 8. From that table we can observe that task inference still happens. The average score for this setting is 50.52, lower than SeededLoRA’s 52.47 average score when using the seed expert,
>
>
> **Q4:  isn’t a trained model already positioned within an optimized subspace? How, then, would additional seeding impact performance? Furthermore, wouldn't a new seed be necessary whenever a new dataset or task is added to the mixture?**
>
> A4: Previous work shows that the spectrum of the Hessian is spanned by a number of large eigenvectors that approximately equivalent to the number of classes in the training set. However, finetuning sets often have very different data. For example, instruction tuning data is very different from pretraining data. This new data leads to optimizations in parameter space where some parameters that were “dormant” become “active” and as such, the new optimization shifts slightly, but significantly, therefore requiring the seeded approach to let the optimization landscape settle in its new local environment.
>
> **Q5: A crucial ablation study would be to evaluate performance on out-of-distribution datasets. [...] This would clarify whether re-seeding is essential for a new dataset of the same task. [..] For a completely new task, however, I would certainly expect that re-seeding would be required, but proving that with an ablation would make the experiment section stronger.**
>
> A5: We also asked ourselves this question, but since we are computationally limited, we trusted in the result from Li et al.[1],  who showed that training a seed model on javascript still achieved local mode connectivity across diverse, unrelated domains such as medical, law, arxiv, or C4. If the same holds for adapters, our method should work even if all expert-specific datasets are not available at the initial training phase.
> [1]: https://arxiv.org/abs/2208.03306
>
>
> **Q6: I am uncertain about the purpose of Section 7, as it seems to add little beyond what was already covered in Seeded LoRA**
>
> A6: We chose to emphasize this to highlight that SeededLoRA is equivalent to well-established results in the literature (multi-head attention and ResNeXt). Other methods, like the closely related MoLoRA do not have this relationship. As such, our approach sits on a rich stream of literature which makes further study quite fruitful, particularly with multi-head attention being used everywhere, and ResNeXt being still superior to vision transformers in many vision problems.

---

> ### Author Response · Authors · 2024-11-21
>
> As the discussion period is nearing its end, we wanted to ask Reviewer ouui if there are any follow-up points we can clarify. We have responded to all concerns raised. If there are no further points of clarification regarding the manuscript, experiments, our fine-tuning procedure, and details about the optimization subspace that makes our method possible, we would kindly ask that reviewer ouui to consider increasing their score to reflect this.

---

> > ### Comment · Reviewer_ouui · 2024-11-21
> > **Follow Up Comments and Questions.**
> >
> > Q1 and Q4. Limited computational resources are a valid limitation; however, without these ablation studies, readers may find it difficult to assess whether the proposed approach would enhance or hinder its application to their specific use cases. I still strongly recommend conducting these ablations to strengthen the results and provide greater clarity.
> >
> > Q3. When comparing the results in Table 1 and Table 8, the primary difference appears to be in the Arithmetic tasks, which seem to account for the majority of the observed increase in average scores. If these tasks were excluded, the overall average improvement might be negligible, but if you decided to only filter arithmetic tasks the difference might be huge. Why do you think this is the case? Could it be that seeded LoRA is inherently biased toward certain task types, such as Arithmetic? If so, what aspects of the method contribute to this bias?
> >
> > Q4. It would be helpful to provide proper citations when referencing existing work, as demonstrated in the response to Q5. Additionally, while I acknowledge your point that fine-tuning datasets may have a different data distribution, considering the vast amount of data used to train LLMs, do you think it's valid to argue that there's a significant input distribution shift during fine-tuning? One could argue instead that the primary shift is task-based—from predicting the next token to answering a specific query—while the input distribution likely remains consistent. Therefore, I find your argument regarding distribution shifts a little unconvincing.

---

### Official Review · Reviewer_R2th · 2024-11-04

**Soundness:** 2
**Presentation:** 2
**Contribution:** 2
**Rating:** 5
**Confidence:** 2

**Summary:**

Seeded LoRA is proposed to improve the performance of LoRA without post-merging fine-tuning. The method can be divided into 2 steps:
1. Random sample 5-10% of SFT training data to train seed LoRA weights.
2. The weights trained in step 1 are used to do task-specific fine-tuning separately to get N expert models.
- At inference time, the inference results from each LoRA module are aggregated through the arithmetical average and added with the original LLM weights to get the final inference results.
Experiments showed that this method achieved better performance when compared with the original LoRA and MOE LoRA with router mechanism.

**Strengths:**

This work proposes Seeded LoRA as a multitask fine-tuning method without post-merging fine-tuning. The method is composed of 2 steps:
1. Random sample 5-10% SFT training data to train a seed LoRA module to be used as common ground for task-specific fine-tuning.
2. Use the LoRA weights from step 1 to fine-tune LLM on multiple tasks separately
- At inference time, the outputs from each LoRA module are aggregated through simple averaging and sent to the decoder without post-tuning to generate the results.
Experiments showed the proposed method is better than the original LoRA or MOE LoRA method with a routing module.

**Weaknesses:**

- Novelty seems to be limited. As is cited in the paper, the work seems to be a simple adoption of the model soup method [1] from whole model fine-tuning to LoRA fine-tuning. In the original paper, multiple model candidates are averaged to generate better model parameters. Here we simply replace the whole LLM weights with the LoRA module weights.
- Experiments seem to be insufficient. For example, the LoRA weights from different experts are averaged arithmetically. Intuitively, for different tasks, the weights of each expert should be different during aggregation. Even without a routing module, a set of learnable weights might be helpful to improve the final inference performance.

[1] Wortsman, Mitchell, et al. "Model soups: averaging weights of multiple fine-tuned models improves accuracy without increasing inference time." International conference on machine learning. PMLR, 2022.

**Questions:**

- The current setup is also connected with MAML (model agnostic meta-learning [2]), which is targeted at learning a set of initialization weights, and for each downstream task, only a few examples will be sufficient to achieve good performance. Instead of random sampling, is it possible to apply the idea of MAML here to find the best seed LoRA weights?

[2]Finn, Chelsea, Pieter Abbeel, and Sergey Levine. "Model-agnostic meta-learning for fast adaptation of deep networks." International conference on machine learning. PMLR, 2017.

---

> ### Author Response · Authors · 2024-11-19
>
> Thank you so much for taking the time to review or work and suggesting improvements. We appreciate it and will try to incorporate as many of them as possible in our final submission. Below, we add responses to some of the questions raised. We hope this additional information provides a complete picture of our work and contributions.
>
> **Q1: The work seems to be a simple adoption of the model soup method**
>
> A1: The model soup method averages the weights of multiple models fine-tuned with different hyperparameter configurations. Even though we also average the weights of our fine-tuned experts, our methods are different. In SeededLoRA each expert is trained on different data using the same hyperparameters. What we show in this work is that 1) importance of the **Seed LoRA** before training expert LoRA adapters, where the Seed LoRA trained on the combined data mix provides higher mode connectivity and better merging results 2) Using the Seeded LoRA approach, we can mitigate task inference by using a simple weight average of fine-tuned experts. 3) Our simple method works best and that other work that tries to do routing work might fail exactly because the optimization landscape is not initialized well.
>
> **Q2: Experiments seem to be insufficient. For example, the LoRA weights from different experts are averaged arithmetically. Intuitively, for different tasks, the weights of each expert should be different during aggregation. Even without a routing module, a set of learnable weights might be helpful to improve the final inference performance**
>
> A2: The main point was that our simple method works best and that other work that tries to do routing work might fail exactly because the optimization landscape is not initialized well. Section 6 shows the failure case of MoLoRA routing, that uses soft-merging (i.e. weighted average using learnable parameters) which is subtle but very important. The difficulty of training a good router that can utilize experts effectively is not specific to MoLoRA. There are many examples in the literature that discuss issues with routing in the Mixture-of-Experts models (SiRA, Zhu et al). Additionally, it has been shown that experts in MoE settings do not tend to specialize on specific tasks (Mixtral of Experts, Q. Jiang et al[1]). While we did not do controlled ablations on different seeds, we did find in our experimentation that seeded LoRA seems to be relatively stable with respect to what seed we use. This is in line with finding in the existing literature (see Branch-train-merge; Li et al.[2]). The advantage of arithmetic average is that it does not require any training (i.e. learnable parameter), therefore provides better flexibility. This is particularly important when a new LoRA expert is trained from the Seed LoRA in which we can just apply averaging without further training.
> [1]: https://arxiv.org/abs/2401.04088
> [2]: https://arxiv.org/abs/2208.03306
>
> **Q3: Is it possible to apply the idea of MAML here to find the best seed LoRA weights?**
>
> A3: The exact data used for training the Seed Expert is not critical. For example, Li et al [1]. showed that training a seed model on javascript still achieved local mode connectivity across diverse, unrelated domains such as medical, law, arxiv, or C4. Theoretically, this is linked to how the eigenvalues of the Hessian are linked to the optimization (such as optimal learning rate, and local near-linear optimization). Please see Ghorbani et al[2], and Gur-Ari et al[3], for more theoretical details. MAML gives good initialization per task, but here we want to have a good initialization for expert LoRA to be successfully merged.
> [1]: https://arxiv.org/abs/2401.04088
> [2]: https://arxiv.org/abs/1901.10159
> [3]: https://arxiv.org/abs/1812.04754

---

> > ### Comment · Reviewer_R2th · 2024-11-22
> > **Thanks for the response**
> >
> > Thanks for the response.
> > - Q1: the response is valid. Basically here the models are trained with same hyper parameters but on different dataset. What if we train with different hyper parameters? Intuitively, each task should have different optimal hyper-parameter setup.
> > - Q2: seems the issue is that the routing module parameters are hard to learn, have we tried to learn a better set of soft merging weights?
> > - Q3: "MAML gives good initialization per task, but here we want to have a good initialization for expert LoRA to be successfully merged". If the task is defined as "merging each lora expert", then they are similar.

---

> ### Author Response · Authors · 2024-11-21
>
> As the discussion period is nearing its end, we wanted to ask Reviewer R2th if there are any follow-up points we can clarify. We have responded to all concerns raised. If there are no further points of clarification regarding the manuscript, our contributions and how our method differs from model soups, our experiments, and how we compare to other approaches (MAML), we would kindly ask that reviewer R2th to consider increasing their score to reflect.

---

### Meta-Review · Area_Chair_DBZm · 2024-12-10

**Metareview:**

The submission presents "Seeded LoRA," a variant of low rank adaptation that works with a seed expert LoRA fine-tuned on a small random subset of data, which is then used to initialize downstream task LoRAs.  The reviewers were unanimous in their opinion that the submission falls short of the acceptance criteria for ICLR.

**Additional Comments On Reviewer Discussion:**

Discussion included the relationship to the model soup method of Wortsman et al., to MAML and other multi-task / continuous learning strategies, depth of experimental results.  Two of three reviewers were inactive during the discussion phase, but Reviewer c3Qw was active and had remaining concerns along the lines of other reviewers at the end of the discussion.

---

### Decision · Program_Chairs · 2025-01-22

Reject